# Improvement of Sexual Function and Sleep Quality in Patients with Atopic Dermatitis Treated with Dupilumab: A Single-Centre Prospective Observational Study

**DOI:** 10.3390/ijerph20031918

**Published:** 2023-01-20

**Authors:** Clara Ureña-Paniego, Trinidad Montero-Vílchez, Raquel Sanabria-de-la-Torre, Alberto Soto-Moreno, Alejandro Molina-Leyva, Salvador Arias-Santiago

**Affiliations:** 1Department of Dermatology, Virgen de las Nieves University Hospital, 18012 Granada, Spain; 2Instituto de Investigación Biosanitaria ibs.GRANADA, 18012 Granada, Spain; 3Cell Production and Tissue Engineering Unit, Virgen de las Nieves University Hospital, Andalusian Network, 18014 Granada, Spain; 4Department of Dermatology, Faculty of Medicine, University of Granada, 18071 Granada, Spain

**Keywords:** atopic dermatitis, dupilumab, sexual health, sleep quality

## Abstract

Atopic dermatitis (AD) is a chronic inflammatory skin disease presenting as xerosis, eczema and intense pruritus. These symptoms negatively impact patients’ quality of life. However, the effect of AD on sexual function and sleep quality and how treatment with dupilumab could modify them have not been explored in depth. The aim of this study is to assess the effects of dupilumab on sexual and sleep quality in patients with AD. For that purpose, an observational prospective study was designed. Patients were evaluated at baseline and after 16 weeks of dupilumab treatment. Disease severity was assessed by Eczema Area and Severity (EASI) and SCORing Atopic Dermatitis index (SCORAD). Sexual function was evaluated using validated questionnaires, for men via the International Index of Erectile Dysfunction 5 (IIEF-5) and for women via the Female Sexual Function Index (FSFI). Sleep impairment was recorded through Pittsburgh Sleep Quality Index (PSQI). Thirty-two patients, with a mean age of 30.53 ± 14.48 years old, were included. Regarding sex, 59.8% (20) were female. Most patients had a severe disease reflected in a mean basal EASI of 23.24 ± 6.74 and a SCORAD of 54.07 ± 13.89. Clinical scores improved after dupilumab treatment. At baseline, 47.37% women presented sexual dysfunction and 66.67% men had erectile dysfunction. FSFI improved from 23.51 to 27.93 points (*p* = 0.008) after dupilumab. Desire, arousal, satisfaction and pain were the components with great improvement. Women with a great improvement in FSFI showed greater clinical results and increased quality of life. At first, 96.9% (31/32) of participants presented with poor sleep quality. After treatment with dupilumab, sleep quality was enhanced and PSQI scores decreased from 12.8 points at baseline to 7.73 points (*p* < 0.001). In conclusion, dupilumab is associated with reduced sexual dysfunction, mainly in women, and sleep quality.

## 1. Introduction

Atopic dermatitis (AD) is a prevalent chronic inflammatory skin disease consisting of the presence of pruriginous and eczematous lesions. It presents with symptoms that greatly impair the patient’s quality of life (QoL), affecting several domains, such as leisure, productivity, personal relationships and sleep quality [1,2,3,4].

Sexual health impairment has not been consistently assessed in patients with AD, despite the societal and personal importance of sex and its key role within patient-reported outcomes and impact on QoL [1,5,6]. It has been observed that AD patients are more likely to report sexual dysfunction than healthy controls [7,8]. The convergence of a pro-inflammatory state, abnormal hormonal levels and the psychological impact of eczema on genital areas might explain the high levels of sexual dysfunction reported by male and female patients with AD [9,10]. Nevertheless, to date, only one report has evaluated the impact of an adequate treatment on sexual dysfunction in patients with AD [5].

Another aspect that contributes to poor QoL in AD patients sleep loss. Sleep disturbance has been linked to AD, and it is one of the most important factors that impairs QoL in AD patients. Thus, patients with AD experience regular fatigue and reduced productivity and concentration during daytime caused by nocturnal awakenings and prolonged sleep induction due of pruritus [4,11]. Disease control is related to better sleep quality. While 23.8% of patients with inadequate control of AD experienced sleep disturbances, only 8.5% did so when it was adequately controlled. The SCORing Atopic Dermatitis index (SCORAD), one of the most widely used indexes for AD severity evaluation contemplates subjective sleep loss among its items, even though the information it provides is deficient and unidimensional [12]. 

The link between sleep and sexual quality is unclear. Conflicting findings point to opposite directions regarding the relationship between both dimensions of QoL [13]. It seems sleep duration is directly related to sexual desire and increases the likelihood of engaging in sexual intercourse, even when it may decrease unstimulated sexual arousal in women. In men, erectile dysfunction can be caused by poor quality of sleep, but the role of testosterone and how it responds to sleep patterns seems to be more important [14,15].

Dupilumab is a fully human monoclonal antibody that acts through the inhibition of interleukin 4 and 13 signaling, key molecules in the pathogenesis of AD, and in other type-2-immunity-driven diseases. Dupilumab is an effective treatment for AD, as it reduces the Investigator Global Assessment (IGA), Eczema Area and Severity (EASI) and SCORAD; and it also improve patients’ QoL as assessed by DLQI [1,16,17]. Nevertheless, the burden of moderate and severe AD is not completely captured in these scenarios. Patient-reported outcomes are receiving increasing attention in the context of AD clinical trials. Thus, specific domains of QoL, such as sleep and sexual quality, should be taken into account. Several authors have explored the effect of dupilumab on sleep quality with remarkable results. It seems dupilumab reduces sleep disturbances even after one week of treatment, and it maintains this effect while treatment persists. Regarding the sexual sphere, less research is available. Napolitano et al. observed an improvement in sexual desire after dupilumab treatment. For both cases, the use of validated and comprehensive measuring tools for both sleep disturbances and sexual function was lacking. Further work is needed to increase the evidence on this topic and to explore the interrelationships between sexual and sleep quality of life and how dupilumab may impact on these things [5,18,19].

The aims of this study were to (1) evaluate the effects of dupilumab on sexual function and sleep quality in patients with AD; (2) to explore the relation among sexual function, sleep disturbances and disease severity; (3) to identify factors related to an improvement in sexual function.

## 2. Materials and Methods

### 2.1. Study Design and Patients

A prospective observational study was conducted, in which participants were recruited from January 2021 to June 2022 in the Dermatology Department, Hospital Universitario Virgen de las Nieves, Granada (Spain).

Eligible patients were adults diagnosed with AD by a dermatologist according to Hanifin and Rajka criteria [20,21], between 18 to 65 years of age, who were starting dupilumab treatment. Clinical infection, a history of cancer, an immunological disease or another inflammatory skin disease, incapacity to comply with the protocol and refusal to participate in the study were exclusion criteria. 

Dupilumab was administered subcutaneously, 600 mg initially as a loading dose, which was followed by weekly administration of 300 mg every other week.

### 2.2. Measures

All measures were recorded at baseline and at week 16 after the start of dupilumab.

AD severity was evaluated by a trained dermatologist by EASI, SCORAD, Dermatology Life Quality Index (DLQI) and IGA. 

In female patients, sexual dysfunction was evaluated via the Female Sexual Function Index (FSFI). The FSFI is a self-reported, clinically validated questionnaire that evaluates all of the six domains of female sexual function: desire, arousal, lubrication, orgasm, satisfaction and pain. Scores range from 2 to 36; results below 26.5 being representative of sexual dysfunction [22]. For male participants, the Five-Item International Index of Erectile Function (IIEF-5) was used for the same purpose. It consists of five questions and contemplates varying degrees of erectile dysfunction (from absent, to mild, moderate and severe erectile dysfunction). Values below 22 indicate some level of sexual dysfunction [23].

Sleep QoL was measured with the Pittsburgh Sleep Quality Index (PSQI). It assesses sleep quality in the past month in a 19-item scale. Scores range from 0 to 21; global scores above 7 identify clinically significant sleep disturbances. It accounts for several sleep components, such as subjective quality of sleep, latency, duration, efficiency, alterations, drug use and daytime dysfunction [24].

All the aforementioned questionnaires have been previously validated in a Spanish population [25,26,27,28,29]. 

Other variables were recorded by a clinical interview and physical examination: sex, age, marital status, smoking habit, alcohol consumption, frequency of topical corticosteroid use, AD family history, disease evolution, atopic symptoms and previous AD treatments.

#### Statistical Analysis

Descriptive statistics were used to present the sample characteristics. Continuous data were expressed as the mean (standard deviation) and qualitative data as relative (absolute) frequency. The Shapiro–Wilk test was used to determine the normality of data distribution and Levene’s test to check the homogeneity of variance. The Student’s *t*-test for paired samples was used to compare differences in parameters before and after dupilumab. The Student’s *t*-test for independent samples was used to compare differences in parameters between different populations. The Pearson correlation test was used to assess the relation between different variables. Statistical significance was defined as a two-tailed *p* < 0.05. SPSS version 24.0 (SPSS Inc., Chicago, IL, USA) was used for statistical analyses.

### 2.3. Ethics

This study was approved by the ethics committee of Hospital Universitario Virgen de las Nieves (HC01/0442-N-20). The nature of the study was explained to all participants, who agreed to participate by giving their verbal and written consent. All measurements were non-invasive, and the confidentiality of participants’ data was strictly preserved.

## 3. Results

No differences in sociodemographic or clinical characteristics were found between men and women; see Appendix A. Thirty-two patients were included in the study, of whom 62.5% (20/32) were female. The mean age was 30.53 ± 14–48 years old, 28.1% (9/32) of the patients had a familiar history of atopic dermatitis and 56.3% (18/32) presented with signs of atopic march. A detailed depiction of demographic and clinical characteristics of the population is shown in Table 1.

### 3.1. Clinical Improvement

Most patients included had severe disease, reflected in a mean basal EASI of 23.24 ± 6.74 and a SCORAD of 54.07 ± 13.89. EASI significantly improved 16.05 points (*p* < 0.001) after 16 weeks of dupilumab treatment. Similarly, mean SCORAD and po-SCORAD improved from 53.81 ± 15.11 to 12.8 ± 2.51 (*p* < 0.001) and from 69.42 ± 16.7 to 34.81 ± 22.13 (*p* < 0.001), respectively (Figure 1). Dupilumab also decreased the IGA score (3.48 ± 0.65) and POEM (23.47 ± 4.82). Additionally, it improved quality of life assessed by DLQI (from 15.75 ± 7.54 to 5.63 ± 1.08, *p* < 0.001).

### 3.2. Sexual Quality of Life

At baseline, 47.37% of women presented sexual dysfunction by the FSFI questionnaire and 66.67% of men had erectile dysfunction assessed by IIEF-5. Overall, an improvement in sexual function was observed after treatment with dupilumab. For women, there was an improvement of 4.38 points in FSFI (from 23.51 to 27.93 points; *p* = 0.008). Nevertheless, not all spheres of female sexual function were affected equally. Desire, arousal and pain were the components that experimented a great improvement after dupilumab (*p* = 0.005, *p* = 0.008 and *p* = 0.007, respectively) (Figure 2). In men, no change was observed in the IIEF-5 score after treatment (from 19.67 to 20.5 points; *p* = 0.5) (Figure 3).

### 3.3. Sleep Quality of Life

Initially, 96.9% (31/32) of participants presented with poor sleep quality. After treatment with dupilumab, sleep quality was enhanced: PSQI scores decreased 5.07 points on average (from 12.8 points at baseline to 7.73 points; *p* < 0.001). In fact, most components of sleep quality were significantly improved, such as subjective sleep quality (*p* < 0.001), sleep latency (*p* < 0.05), duration (*p* < 0.05), efficiency (*p* < 0.001) and daytime dysfunction (*p* < 0.05). Only sleep alteration and drug use rendered non-significant differences in the score after treatment with dupilumab (Figure 4).

### 3.4. Relationship among AD Severity, Sexual Function and Sleep Quality

There was a negative correlation between IIEF-5 results and SCORAD values after treatment, meaning that a reduction in SCORAD was accompanied by an improvement in sexual function in males (r = −0.88; *p* = 0.008). Additionally, final FSFI scores negatively correlated with the final PSQI, implying greater improvement in sexual quality of life associated with better sleep quality (r = −0.51; *p* = 0.027). Moreover, women who improved more than four points on FSFI also improved more than their counterparts in SCORAD (−35.36 vs. −19.68; <0.09), POEM (−18.75 vs. −11.9; *p* < 0.035) and quality of life DLQI (−18.76 vs. −4; *p* < 0.001).

Individual responses of the main studied variables can be found in Appendix A.

## 4. Discussion

This study evaluated the effects of dupilumab on clinical and patient-oriented outcomes, particularly sleep quality and sexual function, through the use of validated questionnaires. Dupilumab improved clinical AD scores and sexual function and sleep quality. An improvement in sexual function in women was related to greater improvement in AD severity and quality of life.

The epidemiology and causes of sexual dysfunction among patients with AD are yet unclear. We observed that more than two thirds of men had erectile dysfunction and almost half of women suffered from sexual dysfunction. In the literature, a large Danish cross-sectional study showed no increased risk of erectile dysfunction in men with AD in comparison with the healthy population and patients with psoriasis [30], whereas a Taiwanese case-control study found men with AD were 1.60 more likely to experience erectile dysfunction [9]. Regarding females, patients with AD suffer from lower self-perception and sexual satisfaction compared to the general population [8]. Moreover, men and women with severe AD reported genital eczema more frequently than those with moderate or mild disease, implying a higher disease burden, lower QoL and decreased sexual desire [6,8,31]. The latter could be explained by sexual wellbeing in women being tightly tied to genital appearance satisfaction [32]. We observed that dupilumab improved female sexual function, but it did not greatly change sexual function in men. Given the effectiveness of dupilumab for genital eczema and the fact that in our population, an improvement of over four points in FSFI was related to more favorable DLQI and EASI scores, the role of genital eczema in female sexual dysfunction should be considered [33]. Results from randomized dupilumab trials, such as SOLO 1 and 2, indicate improvements in sexual function within the DLQI questionnaire by more than 30% compared to placebo users [1], but the study did not explore the difference in effect between men and women. Similarly, a lack of differences between males and females regarding improvement in sexual function was observed by Linares et al. [5]. More studies are needed to know if the impact of dupilumab in sexual function actually differs between the sexes.

A drastic improvement in AD patients’ sleep quality with dupilumab has already been reported by several authors [1,3,16]. The cause of sleep disturbance in these patients is complex, and the pathogenesis remains obscure. There is a relationship between sleep AD severity and sleep disturbances [34,35]. The literature points to several factors contributing to poor sleep quality in patients with AD, such as scratching, asthma nocturnal symptoms and learned insomnia [34,36,37]. Polysomnographic studies show patients with AD present with disruption in rapid eye movement (REM) and non-rapid eye movement (NREM) phases [34]. Elevated levels of various biomarkers that can potentially lead to impaired sleep, such as acetylcholine, norepinephrine, histamine and IL-6, have been reported in AD patients [36,38]. It seems AD alters the circadian rhythm through abnormal cytokine production and melatonin dysregulation during the nighttime, leading to fragmented sleep [34]. Multiple authors have emphasized the role of melatonin in AD-related sleep disorders. Melatonin is not only produced in the pineal gland; the skin, lymphocytes and mast cells are important melatonin sources. Serum melatonin is decreased in patients with AD, and its levels correlate with disease severity. Melatonin supplementation can lead to improvements in SCORAD and in sleep parameters. Examining the effect of melatonin on pruritus, however, has yielded contradictory results [39,40]. At the same time, it is quite likely that sleep disturbance exacerbates AD symptoms by the inflammatory effect of sleep deprivation and the chronic stress poor sleep quality is associated with. The reason behind the better subjective sleep quality in AD patients treated with dupilumab could be related to how effective dupilumab is in pruritus reduction. In fact, dupilumab improves pruritus as early as after 2 days of treatment, which is shortly followed by improved sleep quality [1,3,41]. Given its primal role in the sleep cycle, dupilumab could exert some effect on melatonin production. There is no evidence in the literature to support this claim, but an ongoing clinical trial exploring the effect of dupilumab on circadian rhythms may shed some light on this topic [42].

There is little research studying the relationship between sleep quality and sexual function within the context of AD. Even though poor sleep quality is associated with increased sexual arousal, it is also related to erectile dysfunction and orgasmic disorders [43]. In healthy individuals, sexually active women report good sleep quality more frequently than sexually inactive women [13,43]. Some authors suggest pruritus is the common factor for poor sleep quality and sexual dysfunction in adults with AD. It is reported that 23.5% of patients considered pruritus to affect negatively their sexual arousal and 87.1% to impact their ability to fall sleep [44,45]. Studies exploring the relationship between sexual function and sleep quality suggest hormonal imbalances may be at the center of it. Impairment of sleep in women due to disorders such as obstructive sleep apnea and insomnia are related to lower levels of progesterone and estradiol. Both hormones have a strong dependence on circadian rhythms and play a role in sexual desire and satisfaction (13). For both sexes, testosterone levels peak during the nighttime, and insufficient or poor sleep can have an adverse impact on them. Levels of testosterone correlate in males and females with sexual desire and frequency of sexual activity. The cross-sectional nature of most of these studies hinders our understanding of the causality between sleep and sexual function [46,47]. There is insufficient research available on the impact of AD treatment on sexual health, in contrast to a growing body of evidence of its effect on sleep quality, especially with novel treatments such as Janus kinase inhibitors and other monoclonal antibodies, such as tralokinumab. In consequence, we can only hint at the hypothesis that dupilumab improves sleep quality, which would have a beneficial effect on sexual quality of life, until more research is performed on the subject. Future studies are needed to assess the impacts of other AD therapies on sleep quality and sexual health.

Dermatologists are responsible for providing holistic care for patients with AD, and given its impact on QoL, sexual health and sleep quality should be assessed in all patients at baseline and as measures of treatment response. The presence of any of these disorders could be considered a criterion of severity when electing treatment. 

There are several limitations to this study, such as the limited sample size, especially regarding the number of men, the lack of a control group and its unicentric nature. Moreover, the questionnaire for females might be more precise and multidimensional than the one used for males. Among the strengths of this study, the most notable are its prospective nature and the use of validated questionnaires.

## 5. Conclusions

Dupilumab may improve sexual and sleep quality in adults with AD. Women might benefit more in the sexual sphere than men. Given the impacts of sleep disturbances and sexual impairment on QoL in patients with AD, its assessment could guide management and evaluate response to treatment. Further research is needed to assess dupilumab’s improvement to sexual quality of life and differences between men and women.

## Figures and Tables

**Figure 1 ijerph-20-01918-f001:**
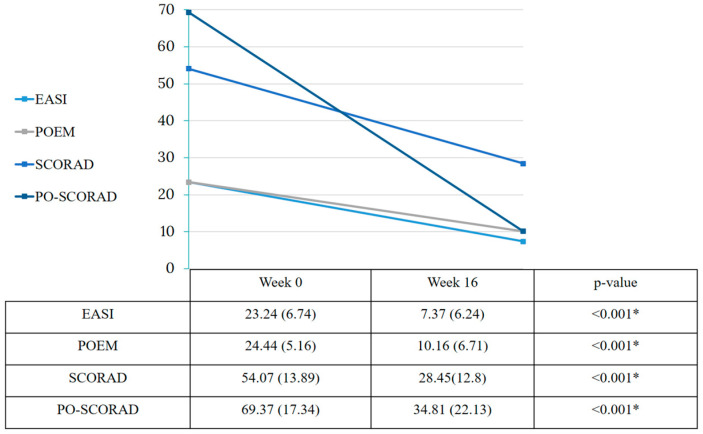
Effect of dupilumab on AD severity in week 0 and 16. Two-tailed * *p* < 0.05 was considered statistically significant in all tests.

**Figure 2 ijerph-20-01918-f002:**
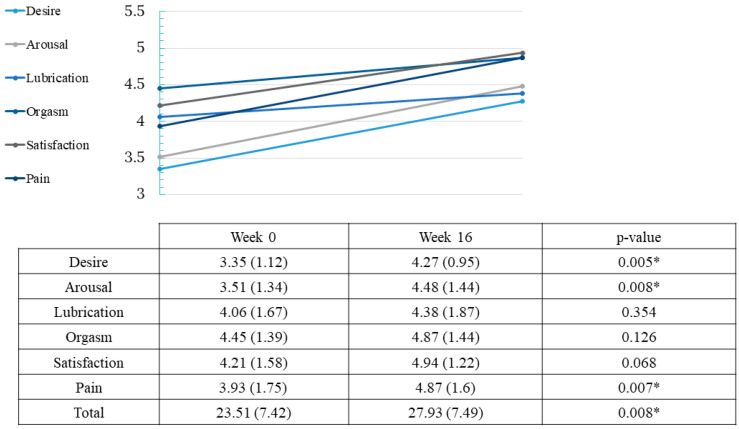
Effect of dupilumab on female sexual function. Values of total FSFI and subcategories in weeks 0 and 16. Two-tailed * *p* < 0.05 was considered statistically significant in all tests.

**Figure 3 ijerph-20-01918-f003:**
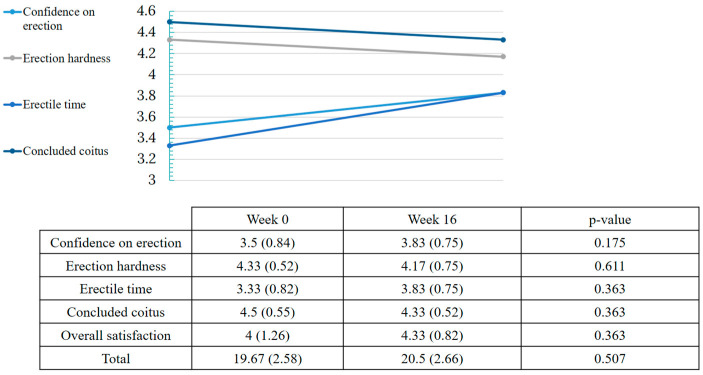
Effect of dupilumab on male sexual function. Values of total IIEF-5 and subcategories in weeks 0 and 16.

**Figure 4 ijerph-20-01918-f004:**
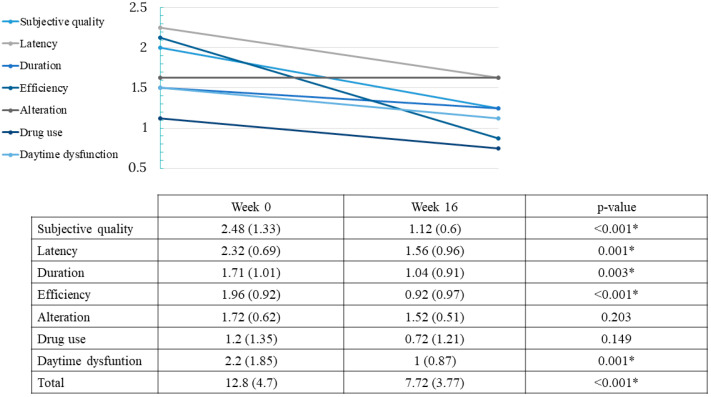
Effect of dupilumab on sleep quality. Values of total PSQI and subcategories on weeks 0 and 16. Two-tailed * *p* < 0.05 was considered statistically significant in all tests.

**Table 1 ijerph-20-01918-t001:** Sociodemographic and clinical characteristics at baseline.

	Total	Male	Female
	(*n* = 32)	(*n* = 12)	(*n*= 20)
Age (years)	31.81 (13.93)	30.71 (19.54)	30.4 (14.48)
Marital status			
Single	68.8% (22)	75% (9)	65% (13)
Married	9.4% (3)	8.3% (1)	10% (2)
Divorced	9.4% (3)	0%	15% (3)
Smoking habit (yes)	9.4% (3)	8.3% (1)	10% (2)
Alcohol consumption (yes)	28.1% (9)	25% (3)	30% (6)
Topical corticosteroid use	65.6% (21)	58.3% (7)	70% (14)
Frequency of topical corticosteroid use (>4 times/week)	24% (8)	16.3% (14)	30% (8)
Family history of AD	28.1% (9)	33.3% (4)	25% (5)
Atopic march symptoms	56.3% (18)	58.3% (7)	55% (11)
Previous AD treatment (yes)	100% (32)	100% (12)	100% (20)
Systemic treatment	93.8% (30)	91.7% (11)	95% (19)
Biologic treatment	3.1 (1)	8.3% (1)	0% (0)

Data are expressed as mean (standard deviation) or relative (absolute) frequency. AD, atopic dermatitis. Compare qualitative variables.

## Data Availability

The data presented in this study are available from the corresponding author on request.

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
