# Peer review of "Improvement of Sexual Function and Sleep Quality in Patients with Atopic Dermatitis Treated with Dupilumab: A Single-Centre Prospective Observational Study"

_ijerph, 2023, doi:10.3390/ijerph20031918_

Round 1
Reviewer 1 Report
Regarding the manuscript entitiled " Improvement of sexual function and sleep qulity in patients with atopic dermatitis treated with dupilumab", my comments are as follows: [Abstract] Patients status such as ages, genders, should be listed [Material and methods] 1. Indications and severity of patients' atopic dermatitis should be included. 2. Patients' ages ranged from 18-65 years seems too wide for the assessment of sexual function as well as sleep qulity. 3. The dosage of dupiumab should be adiministrated based on patient's body weight. [Result] 1. Based on the range of ages, the results of sexual function in Figiuren 2, Figure 3 should be revised. 2. Sleep qulity of life has to be revised due to wide range of patients' age in Figure 4. [Discussion] 1. The authors need to discuss more about the relationship between sleep qulity and sexual dysfunction due to atopic dermatitis. 2. Since the limited sample size and wide range of patients' ages, and no control group for comparison the effectiveness of dupilumab, it is hard to make the conclusion. More data is needed. 3.The pillfalls can be found in the comparison of sexual function in comparison of male and female as well. In conclusion, the manuscript is interesting. However, more data should be included for making the solid conclusion.Author Response
Regarding the manuscript entitiled " Improvement of sexual function and sleep qulity in patients with atopic dermatitis treated with dupilumab", my comments are as follows:
[Abstract] Patients status such as ages, genders, should be listed
Thank you for the comments. We have incorporated such information in the abstract.
[Material and methods]
- Indications and severity of patients' atopic dermatitis should be included.
We have accordingly included the suggestion mentioned above.
- Patients' ages ranged from 18-65 years seems too wide for the assessment of sexual function as well as sleep quality.
Thank you for the comment. We have assessed sexual function and sleep quality in adults’ patients with AD and have only excluded children and the elderly, as previous reports (Cork, 2020; Kong, 2016). Moreover, Sampogna et al. (2017) found no differences in sexual dysfunction between different age groups in participants below 60 years old. In that way we don’t think the age range could impact in our results.
- The dosage of dupilumab should be adiministrated based on patient's body weight.
Thank you for the comment. We administered dupilumab following the current guidelines for atopic dermatitis (doi: 10.1111/jdv.18345) and the Spanish Agency of Medicines and Medical Devices (AEMPS) (https://cima.aemps.es/cima/dochtml/p/1171229006/P_1171229006.html).
[Result]
- Based on the range of ages, the results of sexual function in Figure 2, Figure 3 should be revised.
- Sleep quality of life has to be revised due to wide range of patients' age in Figure 4.
Thank you for both suggestions. It would be interesting to carry out further studies with more participants to evaluate if age could have an impact on sexual function and sleep quality in patients with AD. Previous reports have no observed differences in sexual dysfunction in participants below 60 years old. Following your recommendation, we have included new supplementary figures with the individual values of EASI, PSQI, FSFI and IIEF-5.
[Discussion]
1.The authors need to discuss more about the relationship between sleep quality and sexual dysfunction due to atopic dermatitis.
We agree with this remark, accordingly, we have revised the Discussion section to emphasize this point.
The following information has been included: Studies exploring the relationship between sexual function and sleep quality suggest hormonal imbalances may be at the center of it. Impairment of sleep in women due to disorders such as obstructive sleep apnea or insomnia are related to lower levels of progesterone and estradiol. Both hormones have a strong dependence on circadian rhythms, playing a role in sexual desire and satisfaction (13). For both sexes, testosterone levels peak during night-time and insufficient or poor sleep can have an adverse impact on them. Levels of testosterone correlate in males and females with sexual de-sire and frequency of sexual activity. The cross-sectional nature of most of these studies hinder our understanding of the causality between sleep and sexual function [38,39]. There is insufficient research available on the impact of AD treatment on sexual health in contrast to a growing body of evidence of its effect on sleep quality, especially with novel treatments like Janus Kinase inhibitors or other monoclonal antibodies as tralokinumab. In consequence, we can only hint at the hypothesis that dupilumab im-proves sleep quality which would have a beneficial effect on sexual quality of life until more research is done on the subject. Future studies are needed to assess the impact of other AD therapies on sleep quality and sexual health.
- Since the limited sample size and wide range of patients' ages, and no control group for comparison the effectiveness of dupilumab, it is hard to make the conclusion. More data is needed.
Thank you for the comment. In this study we have observed that patients treated with dupilumab improved sexual function and sleep quality comparing baseline values and 16-weeks after treatment. This is an exploratory study and does not intend to lay the foundations on this topic. Our intention with this publication is to show other researchers our findings to bring attention to this topic and encourage its discussion. It would be interesting to carry out further studies comparing dupilumab with other treatments including ciclosporin or JAKi. Following your recommendations, we have included this information as a limitation of our study.
- The pitfalls can be found in the comparison of sexual function in comparison of male and female as well.
Thank you for the comment. We haven’t compared sexual function between males and females. We have evaluated how dupilumab changes sexual function in males and females. To assess sexual function in female patients we used the Female Sexual Function Index (FSFI), a self-reported clinically validated questionnaire that evaluates all of the six domains of female sexual functions: desire, arousal, lubrication, orgasm, sat-isfaction and pain. Scores range from 2 to 36, being results below 26.5 representative of sexual dysfunction [20]. For male participants, the Five-Item International Index of Erectile Function (IIEF-5) was used for the same purpose. It consists of five questions and contemplates varying degrees of erectile dysfunction (from absent, to mild, mod-erate and severe erectile dysfunction). Values below 22 indicate some level of sexual dysfunction [21]. We agree that the questionnaires are different but they are validated questionnaires that have previously used to assess sexual function (Linares-González, 2022). Following your recommendations, we have included this comment as a limitation of our study.
Reviewer 2 Report
The authors present an interesting study about the specific aspects of sleep disturbance and impairment of sexual aspects in quality of life in the context of therapeutic response to IL4/IL13 receptor blockade (dupilumab) in patients with moderate-severe atopic dermatitis.
These outcomes are important aspects of therapeutic response and the questionnaires used have not been used in previous larger studies. Therefore, this study adds important information. Unfortunately, the study is relatively small and monocentric, limiting the reliability of the results. Especially the group of men in the study is small (n=12), the high number of reported erectile dysfunction is remarkable but might be incidental.
The article includes a well written introduction and thoroughly describes materials and methods. The results part requires some improvement as outlined further on. The discussion is written very clearly and has a good structure, I am missing some aspects, though.
I would suggest some modifications to further improve the manuscript.
Minor remarks:
- The title should include the nature of the study: “monocentric prospective observational study”, the abstract should include the nature of the study and the number of patients included (n=32)
- in the introduction, some references are written in bold letters, please correct
- I would suggest graphs illustrating the treatment response with each patient representing dots at two given points of time (Figures 1-4). This would allow to comprehend individual therapeutic responses rather than an average response within the study group. If the authors disagree, it might be considered as online supplement. The point that I am making is that the shown graph contains no information on outliers. Besides, were there any interesting findings regarding subgroups? At first the study population is subdivided into different categories, but after that no information is provided regarding these groups.
- discussion: I am missing some comment on potential specific effects of dupilumab rather than effects of improved AD in general (improvement associated with improvement of EASI / SCORAD generally associated with better sleep and sexual well-being?). Would you expect similar findings with other highly efficient AD treatments such as tralokinumab or JAKi? Is there any data on conventional systemic treatments such as CsA?
- I might be wrong but I had the feeling that the questionnaire for females was more precise and multidimensional than the one used for males. Erectile function is not the only dimension of male sexuality, can you draw any conclusions on arousal and desire in your study population in the male-group?
Author Response
The authors present an interesting study about the specific aspects of sleep disturbance and impairment of sexual aspects in quality of life in the context of therapeutic response to IL4/IL13 receptor blockade (dupilumab) in patients with moderate-severe atopic dermatitis.
These outcomes are important aspects of therapeutic response and the questionnaires used have not been used in previous larger studies. Therefore, this study adds important information. Unfortunately, the study is relatively small and monocentric, limiting the reliability of the results. Especially the group of men in the study is small (n=12), the high number of reported erectile dysfunction is remarkable but might be incidental.
The article includes a well written introduction and thoroughly describes materials and methods. The results part requires some improvement as outlined further on. The discussion is written very clearly and has a good structure, I am missing some aspects, though.
I would suggest some modifications to further improve the manuscript.
Minor remarks:
- The title should include the nature of the study: “monocentric prospective observational study”, the abstract should include the nature of the study and the number of patients included (n=32)
We have modified the title following your recommendations: Improvement of sexual function and sleep quality in patients with atopic dermatitis treated with dupilumab: A monocentric prospective observational study
- in the introduction, some references are written in bold letters, please correct
Changes have been made accordingly.
- I would suggest graphs illustrating the treatment response with each patient representing dots at two given points of time (Figures 1-4). This would allow to comprehend individual therapeutic responses rather than an average response within the study group. If the authors disagree, it might be considered as online supplement. The point that I am making is that the shown graph contains no information on outliers. Besides, were there any interesting findings regarding subgroups? At first the study population is subdivided into different categories, but after that no information is provided regarding these groups.
We agree with this aspect and have incorporated such figures as supplementary material. We have not performed subgroups analysis due to the limited sample size and because previous studies found no differences in sexual dysfunction between different age groups in participants below 60 years old (Sampogna et al., 2017). It would be interesting to explore your suggestion in future studies with larger sample size.
- discussion: I am missing some comment on potential specific effects of dupilumab rather than effects of improved AD in general (improvement associated with improvement of EASI / SCORAD generally associated with better sleep and sexual well-being?). Would you expect similar findings with other highly efficient AD treatments such as tralokinumab or JAKi? Is there any data on conventional systemic treatments such as CsA?
Thank you for this interesting remark. Sleep quality has been indirectly studied among AD patients treated with Janus Kinase Inhibitors and Monoclonal antibodies (Blauvelt, 2022; Cork, 2020; Farnam, 2022). On the other hand, evidence about the improvement on the sexual sphere is scarce. We will carry out further studies comparing the effect of dupilumab with other treatments including cyclosporin and JAKi. Improvement in sexual and sleep quality could be in part due to a reduction in disease severity, especially itch intensity but we cannot affirm other novel treatments could be as effective in both domains. It would be interesting to design a new study comparing the effect of tralokinumab or JAKi on the aforementioned domains in the future.
- I might be wrong but I had the feeling that the questionnaire for females was more precise and multidimensional than the one used for males. Erectile function is not the only dimension of male sexuality, can you draw any conclusions on arousal and desire in your study population in the male-group?
Thank you for pointing this out. Truly, disparities between both questionnaires are remarkable. Nevertheless, both IIEF and FSFI are validated scores for exploring sexual dysfunction in both males and females respectively and are in fact widely used in the literature, included when exploring the impact of inflammatory dermatosis on quality of life (Kędra, 2022; Linares, 2021; Varney, 2022). Following your recommendations, we have included this information in the limitation section.
Round 2
Reviewer 1 Report
In response to your asking, my comments are as follows:
[Abstract]
Please explain more about the revised sentence “32 patients, with a mean age of 30.53+_14-48year old”.
[Materials and Methods]
1. Why the authors put the results of Basal EASI and SCORAD in M&M for revision ?
2. Since the 32 patients (20 females v.s. 12 males) can not be calculated and compared for statistical analysis. How did the authors make it!
[Results]
From Table 1, it showed difference between male and female patients regarding topical steroid use and biologic treatment. They didn’t do any comparison in EASI, POEM, SCORAD, PO-SCORAD.
[Discussion]
The sleep disorder related to many factors or etiologies such as ulcers (physical), asthma (medical), depression and anxiety disorders (psychiatric), alcohol consumption, biological clocks, genetics, medications, and aging. The authors must mention in the Introduction or do analyze and discuss for making their hypothesis more solid.
[Conclusion]
Limited samples and uneven gender numbers make strong conclusions hardly.
In conclusion, this manuscript didn’t make adequate revision. Many data still needed !
Author Response
In response to your asking, my comments are as follows:
Thank you for the comments
[Abstract]
Please explain more about the revised sentence “32 patients, with a mean age of 30.53+_14-48year old”.
This was a typographical mistake. We meant 30.53 ± 14.48. It has been checked.
[Materials and Methods]
- Why the authors put the results of Basal EASI and SCORAD in M&M for revision ?
In the previous revision we understood that reviewers wanted that we included the basal EASI and SCORAD in the material and method section and we included it in material and method section and results section. We have now deleted it in the material and method section and have only mentioned it in the result section.
- Since the 32 patients (20 females v.s. 12 males) can not be calculated and compared for statistical analysis. How did the authors make it!
We have not compared female vs males in our manuscript. We have evaluated how dupilumab impact on clinical scores, quality of life and sexual function in our population. The only differences between men and women is that the questionnaire used to evaluate sexual function is different between men and women. Following your recommendations, we have now included a supplementary table comparing sociodemographic and clinical scores between males and females, except for sexual function questionnaire because it is different for men and women and up to our knowledge we have not found any questionnaire that assess sexual function with a similar punctuation in men and women.
[Results]
From Table 1, it showed difference between male and female patients regarding topical steroid use and biologic treatment. They didn’t do any comparison in EASI, POEM, SCORAD, PO-SCORAD.
Following your recommendations, we have now added a supplementary table comparing these scores between men and women. No statistical differences were found between the basal demographic and clinical characteristics neither regarding the clinical response between both genders.
[Discussion]
The sleep disorder related to many factors or etiologies such as ulcers (physical), asthma (medical), depression and anxiety disorders (psychiatric), alcohol consumption, biological clocks, genetics, medications, and aging. The authors must mention in the Introduction or do analyze and discuss for making their hypothesis more solid.
Thank you for pointing this out. The suggested correction has been made on the Discussion section where the change can be found.
The following information has been included: A drastic improvement in AD patients’ sleep quality with dupilumab has already been reported by several authors. The cause of sleep disturbance in these patients is complex and the pathogenesis remains obscure. There is a positive correlation between AD severity and sleep disturbances. The literature points to several factors contributing to poor sleep quality in patients with AD such as scratching, asthma nocturnal symptoms or learned insomnia. Polysomnographic studies show patients with AD present with disruption of rapid eye movement (REM) and non-rapid eye movement (NREM) phases. Elevation of different biomarkers that can potentially lead to impaired sleep such as acetylcholine, norepinephrine, histamine and IL-6 has been reported in AD patients. It seems AD alters circadian rhythm with abnormal cytokine production and melatonin dysregulation during night-time, leading to fragmented sleep. Multiple authors emphasize the role of melatonin in AD relate sleep disorders. Melatonin is not only produced in the pineal gland but the skin, lymphocytes and mast cells are important melatonin sources. Serum melatonin is decreased in patients with AD and its levels correlate with disease severity. Besides, melatonin supplementation can lead to improvement in SCORAD and in sleep parameters. The effect of melatonin on pruritus, however, has yielded contradictory results. At the same time, it is quite likely that sleep disturbance exacerbates AD symptoms by the inflammatory effect of sleep deprivation and the chronic stress poor sleep quality is associated to. The reason behind better subjective sleep quality in AD patients treated with dupilumab could be related how effective dupilumab is in pruritus reduction. In fact, dupilumab improves pruritus as early as after 2 days of treatment shortly followed by sleep quality. Given its primal role in sleep cycle, dupilumab could exert some effect on melatonin production. There is no evidence in the literature to support this claim but an ongoing clinical trial exploring the effect of dupilumab on circadian rhythms may shed some light on this topic.
[Conclusion]
Limited samples and uneven gender numbers make strong conclusions hardly.
Thank you for the comment. We have included them as limitations of our study. We have modified our conclusion, have hypnotized about them and have suggested that more studies are needed to reach solid conclusions: Dupilumab may improve sexual and sleep quality in adults with AD. Women might benefit more in the sexual sphere than men. Given the impact of sleep disturbances and sexual impairment on QoL in patients with AD, its assessment could guide manage-ment and evaluate response to treatment. Further research is needed to assess dupi-lumab's improvement in sexual quality of life and differences between men and women.
In conclusion, this manuscript didn’t make adequate revision. Many data still needed !
Sorry for the inconvenience caused. We expect that this new version has improved the manuscript and respond to all your suggestions. Thank you for the comments.